# Antimicrobial Activity of Graphene-Based Nanocomposites: Synthesis, Characterization, and Their Applications for Human Welfare

**DOI:** 10.3390/nano12224002

**Published:** 2022-11-14

**Authors:** Varish Ahmad, Mohammad Omaish Ansari

**Affiliations:** 1Health Information Technology Department, The Applied College, King Abdulaziz University, Jeddah 21589, Saudi Arabia; 2Centre of Artificial Intelligence for Precision Medicines, King Abdulaziz University, Jeddah 21589, Saudi Arabia; 3Center of Nanotechnology, King Abdulaziz University, Jeddah 21589, Saudi Arabia

**Keywords:** graphene, nanomaterial, antibacterial, antifungal, toxicities

## Abstract

Graphene (GN)-related nanomaterials such as graphene oxide, reduced graphene oxide, quantum dots, etc., and their composites have attracted significant interest owing to their efficient antimicrobial properties and thus newer GN-based composites are being readily developed, characterized, and explored for clinical applications by scientists worldwide. The GN offers excellent surface properties, i.e., a large surface area, pH sensitivity, and significant biocompatibility with the biological system. In recent years, GN has found applications in tissue engineering owing to its impressive stiffness, mechanical strength, electrical conductivity, and the ability to innovate in two-dimensional (2D) and three-dimensional (3D) design. It also offers a photothermic effect that potentiates the targeted killing of cells via physicochemical interactions. It is generally synthesized by physical and chemical methods and is characterized by modern and sophisticated analytical techniques such as NMR, Raman spectroscopy, electron microscopy, etc. A lot of reports show the successful conjugation of GN with existing repurposed drugs, which improves their therapeutic efficacy against many microbial infections and also its potential application in drug delivery. Thus, in this review, the antimicrobial potentialities of GN-based nanomaterials, their synthesis, and their toxicities in biological systems are discussed.

## 1. Introduction

A major issue in the healthcare field is the role of pathogens as infection-causing agents. Pathogens such as bacteria and fungi are frequently reported to be resistant to antimicrobial substances, and thus can undermine the effectiveness of currently used chemicals or antibiotics to treat microbial diseases [1,2]. Numerous molecular and cellular pathways are operated among the microbes for the development of resistance against antibiotics. The major antibiotics-resistant mechanism described are, the P-glycoprotein-mediated efflux of drug, the development of a resistance to absorption or penetration into pathogen cells, the inactivation of therapeutic agents by enzymatic microbial metabolism, endospore/biofilm formation, the mutation/change or protection of drug targets. Thus, there is a great need to search for newer effective therapeutic molecules and to restore the efficacy of treatments [3,4].

In the last few decades carbon-based nanomaterials have drawn the attention of researchers worldwide owing to their low cost, ease of synthesis, and biocompatibility. Among different carbon materials, graphene-based nanomaterials (GBNs) have been well explored over the past decade owing to their unique properties, such as a high surface-to-volume ratio, mechanical flexibility, and a high thermal stability. GBNs show promise in catalysis, solar cells, biosensors, drug delivery, genetic delivery, imaging, photothermal treatment, tissue engineering, and stem cell technology [5,6]. Moreover, they have also been tested in photocatalysis, microbial sensing, and biomarkers sensing [7,8,9,10,11,12]. The potential applications of GBNs are shown in Figure 1.

In addition, GBNs have been well established to have antimicrobial activities and are toxic to both Gram-positive and Gram-negative bacteria. The graphene oxide (GO) and reduced graphene oxide (rGO) are potential inhibitors of bacterial and fungal pathogens; however, the antifungal and antibacterial uses of GBNs are still relatively novel. Over the past decade, interest in GBNs has increased dramatically. Many updates on GBNs are available in different ways that describe the significance of graphene (GN) [7,8,9,10,11,12,13,14,15]. In this review, recent advances in the understanding of their antimicrobial potential, focusing mainly on the antifungal and antibacterial significance of GBNs as nanotherapeutics, have been summarized. In the first part of this review, the synthesis methodology for GBNs is described. The second part summarizes their antimicrobial activities against bacteria and fungi and their applications in microbial diseases, by exploring the therapeutic area of GBNs. The last section includes toxicity studies and future prospects of GBNs [7,8,9,10,11,12,13].

## 2. The Graphene Family and Its Nanoconstruction

GN materials are being investigated for multiple microbiological applications because of their unique physicochemical characteristics. GBNs can be defined and classified according to their characteristics, i.e., morphology, composition, side dimensions, and number of stacked layers [5,6,7]. They are also determined by their carbon/oxygen atomic ratio, material size, degree of deformity, density, surface area, as well as by their flexibility, i.e., bending characteristics [5,6,15,16]. Depending on the method used, GN can be fabricated into various morphologies such as sheets, platelets, ribbons, and quantum dots. Namely, GBNs can be GN, GO, rGO, ultrafine graphite between 5–10 sheets and less than 20 nm in diameter, GN ribbons, GN quantum dots (GQDs), and pristine GN. GN is a 2D sheet of sp2-bonded carbon monolayer and made out of graphite or other carbon sources via the Scotch tape technique, chemical exfoliation, chemical vapor deposition, arc extraction, and carbide phase decomposition. The GO structure consists of single-layer carbon atoms with carboxylate groups in the surrounding environment, where they provide a particular pH based on negative localization, which helps to impart its colloidal stability. GN and GO can be a single to few layered structures and are produced from the reaction of crystalline graphite by variety of methods such as by oxidation of graphite, by sonication, etc. They also contain epoxy, hydroxyl, and carboxylic acid groups on their surface as well as edges. rGO can be synthesized from GO through a degenerative state that includes a high-temperature heat treatment and chemical reactions with hydrazine (N_2_H_4_) or other reducing agents [16,17,18,19].

GQDs are small GN fragments of size less than 1–10 nm and can be synthesized by a variety of methods such as oxidative cleavage, hydrothermal or solvothermal methods, microwave-assisted process/ultrasonic-assisted process, electrochemical oxidation, and carbonization [20,21]. Due to a difference in sizes, number of sheets, and functionalities, each GBN can exhibit unique physiological, morphological, and chemical properties and thus can be used for different desired applications [5,6] (Figure 2). Owing to these unique properties, GN, GO, and other GBNs have been extensively investigated for their therapeutic properties. The common toxicities of cancers or microbial cells are due to the induction of oxidative stress, protein dysfunction, membrane damage, and transcriptional binding. The production of reactive oxygen species (ROS) is a major cause of nanomaterial toxicity [22,23]. The antioxidant enzymes, such as glutathione peroxidase and superoxide dismutase, can reduce or eliminate ROS generation. The disruption in the balance of enzymes, proteins, deoxyribonucleic acid (DNA), and lipids are the key components for cellular death. Furthermore, GO and rGO nanosheets have been reported to have Fenton catalytic activity [9]. Moreover, a structural link between a GBN and its redox function supports the ability to produce ROS from a GBN. Microbial membrane damage is another possible result of hydrophobic interactions between a GBN and membrane phospholipids corresponding to the GBN’s size [24]. Although protein dysfunction and transcription factors are generally not attributed to GBNs, they sometimes contribute to drug delivery and antimicrobial activity [8,9,10]. Investigations have shown that GN–Fe_3_O_4_ leads to *E. coli* protein aggregation. Meanwhile, Fe_3_O_4_ mediate the oxidative properties which cause the breakdown of small proteins themselves. These results suggest that protein dysfunction is much higher in GN–Fe_3_O_4_ in comparison to single or isolated constituent. Due to their stacked layered morphology, GBNs can interact with DNA in a number of groups. For example, the presence of GO next to Cu^2+^ can affect DNA fragmentation by converting Cu^2+^ ions into active oxygen groups in GO nanosheets. GO nanosheets of large size have been reported to produce a significant decrease in antimicrobial performance when tested against *E. coli* (40 mg/mL, 2 h) in comparison to small nanosheets [5,6,25,26]. The detailed descriptions of different types of GBNs and their antimicrobial potentials are described in the antimicrobial section of this manuscript. The exploration of the two-dimensional (2D) nature of GBNs imparted a significant contribution in the field of nanomedicines. Thus, the term “graphene” is generally referred to as atomically thin films. GN and GBNs have been utilized to control microbial pathogens and cancers and have also been significantly applied in many other biomedical applications. Consequently, GO, rGO nanoforms of carbon, and their biomedical potentialities are in common use [5,6,7,8,9]. The temperature-mediated killing of cancer cells under the hyperthermic effect on a nanomaterial is due to the excellent absorption of radiation in the near-infrared. The modification of GBNs with desired moieties can make them uniquely effective in targeting dangerous cancer or tumor cells without harming healthy cells. Nanotechnology involves the use of nanomaterials with different properties compared to many materials with similar structures. GO, due to its large surface area and good functional properties, is gaining valuable attention due to its resistance to chemicals, anticorrosion, and mechanical efficiency [5,6,7,22,23]. The stacked-layered structure, size of sheets, and defects in the GN layers are the major factors that affect its characteristics. GBNs have different characteristics properties which makes them suitable for diverse applications including sensing and drug delivery [7,8,9,10,11,12]. In addition, due to its unique combination of electrical, thermal, physical, and chemical properties, GO has attracted significant interest in a wide range of fields, including biomedical and antibacterial applications [26,27,28,29,30,31]. Other than therapeutics and diagnostics, it has also been used in the storage of electrochemical energy [32]. The properties of GN materials for antibacterial activities are represented in Figure 3.

This article describes the latest advances in biomedical devices of GN–based composite with metal, metal oxide, and polymers and their antimicrobial properties. GN and its composites offer a variety of features such as high stability in the body, antimicrobial properties, good coordination, informal changes, and a multifunction behavior. It is clear that with these important functional properties, GN has proven its potential and won the favor of many biomedical scientists to combat potential pathogen-related disorders. It is now utilized to design newer materials or composite materials as well as to cover or coat materials used in biomedical equipment [5,6,21]. The pi interaction facilitates the loading of zinc phthalocyanine and doxorubicin into the GN basal plane. GN nanosets made by hydrothermal methodology in an alkaline state have potential antibacterial properties against *Enterococcus faecalis* (*E. faecalis*), *Salmonella typhimuirum* (*S. typhimuirum*), *Escherichia coli* (*E. coli*), and *Bacillus subtilis* (*B. subtilis*). This review outline summarizes these promising antimicrobial properties [1,2,3,4,16].

## 3. Properties and Limitations of Graphene Nanomaterials

Various nanoshapes and structures of carbon have been used diversely for human welfare. The other nanometals, such as copper, iron, zinc, and silver, have also been extensively studied for their antimicrobial properties. However, these materials have solubility, stability, dispensability, and more important, toxicity issues [7,25,33]. In contrast, GN is reported to be much safer with excellent properties for antimicrobial action and other applications in biological systems [29]. The most commonly described nanomaterials of carbon are carbon nanofibers (CNF), carbon nanotubes (CNTs), GN nanoplatelets, and rGO [1,2,3,4,5]. Each nanoform of GN offers variable properties and has diverse applications. The unique desirable properties that are required for antimicrobial activity and other biomedical applications are shown in Figure 3. A better colloidal stability is also one of the significant characteristics of GN, which is due to the hydrogen bonds facilitating edged groups or polar oxidizable groups, and the property decreases proportionally with the number of groups, while dispensability is one of the challenges for GN-based materials for biological systems [4,5,16,34,35,36,37,38]. 

The antimicrobial potential of these materials, either alone or in conjugated forms, has been reported [39,40,41] against many pathogens including bacteria, fungi, and viruses [1,2,3,4,42,43,44,45,46].These materials have also been evaluated as safe preservatives to replace the toxic effects of commonly used chemical preservatives [33,34]. Moreover, the synergistic antimicrobial action of GN-based nanocomposites, such as Ag–ZnO-rGO hybrids, with biodegradable polymers, such as polyethylene glycol and agar, has been evaluated in food preparation. The rGO–ZnO hybrid nanomaterial conjugated with the biodegrade polymer PHBV has also been reported to have significant bactericidal activities [1,2,3,4,16,34]. The family of GN offers a specific surface area, biocompatibility, good diffraction strength, high Young’s modulus, fast ionic migration, and high electrical and thermal conductivities. Although carbon nanomaterials, including nanomaterials from the GN family, have excellent surface qualities, they interact with strong interplanar interactions, which makes them insoluble due to the aggregation of particles and thereby limit their antimicrobial properties. Thus, solubility, stability, and antimicrobial action have been reported to increase by reducing the aggregation using nanocarbon-polymer-based hybrid dispersion systems [16,32,37,47].

## 4. Methods of Synthesis of Graphene Nanomaterials

Many methods are used to make the diverse class of GN-based nanomaterials. The main methods that are used in the synthesis of GN nanocomplexes are shown in Figure 4. The synthesis of GN is obtained by various methods including a peel-off methodology, chemical vapor deposition, or by a chemical methodology [17,18,19,20,39,40,41]. The peeling methodology involves separating layers of GN from graphite chunks using adhesive tapes [48], while the chemical vapor deposition involves a pyrolysis of materials to form carbon and a further linking of carbon to form a GN skeleton [49]. This form of GN is obtained as a film or can be freely suspended in solvents. Another route is the synthesis of GO and its subsequent reduction. GO is synthesized by the oxidation of graphite, and the subsequent reduction of GO yields rGO. The reduction of GO into rGO increases the conductivity due to the loss of oxygen content and the formation of a conjugated network [50]. The first reported methodology for the synthesis of GN was the Scotch tape peeling methodology [51]. Later, many other techniques were developed to produce bulk quantities of GN at low cost as per the desired applications. The preferred routes are top-down, bottom-up, and epitaxial growth [52]. Here, some of the most commonly used synthesis methodologies for GN preparation has been summarized.

### 4.1. Exfoliation Methodology

Different methods are used for the synthesis of GN [48,49,50,51,52,53]. The exfoliation methodology means separating a single or a few layers of GN sheets from graphite chunks. The pioneer work of Andre Geim and Konstantin Novoselov used Scotch tape to peel amicrometer-sized GN sheet from staked graphite [54]. The single layer GN sheet was almost transparent with an optical absorption of 2.5% and the approximate estimation of the number of layers could be made by measuring the optical absorbance. Apart from this, Raman spectroscopy, scanning and transmission electron microscopy, and atomic force microscopy were also used for the estimation of the number of layers [55,56]. Other techniques such as photoexfoliation, anodic bonding, and laser ablation are also examples of dry exfoliation methodology. Photoexfoliation uses radiation exposure on graphite sources for the exfoliation process. The report of Lin et al. [57] showed a successful synthesis of thin fluorinated GN nanosheets by UV irradiation of fluorinated graphite. The prepared thin fluorinated GN nanosheets were much thinner than nanosheets conventionally prepared by sonication. The anodic bonding methodology involves the splitting of the GN precursor into few-layered GN sheets upon the application of an electrostatic field. The advantage of this method is its simplicity, and large dimensions of GN sheets up to 100 µm on an insulating substrate can be prepared by this methodology [58]. The laser ablation methodology is similar to the photoexfoliation but here, laser energy is used for the splitting of the GN sheets. The process can be done in liquid phase, in a dry environment, or under specific gaseous conditions [59]. The type of laser and its density is important in controlling the thickness and quality of the generated GN. Apart from this, specific features such as porosity can be incorporated into the sheets by this methodology [60]. The liquid phase exfoliation is the most preferred route for laboratory-scale synthesis. In this process, first, GO is prepared, and its further reduction gives reduced GO commonly known as rGO. rGO shows similar properties as those of GN but is less conductive, with defects in the lattice owing to the oxidation by harsh chemicals and the later removal of oxygen during reduction. The Hummer’s method uses concentrated H_2_SO_4_, NaNO_3_, and KMnO_4_ for the oxidation of graphite into GO [61]. However, the generation of toxic gases such as NO_2_, N_2_O_4_, and ClO_2_ and an uncontrolled temperature rise are the major demerits of this methodology. The modified or improved versions of Hummer’s method uses H_2_SO_4_, KNO_3_, and KMnO_4_, while Tour’s method uses H_2_SO_4_, H_3_PO_4_, and KMnO_4_. The improved and modified Hummer’s method versions eliminate the chances of explosion, and thus, are the preferred techniques for GO synthesis. In Tour’s method, the reaction is done at 50 °C and in its modified version, room temperature reactions have been reported [62]. The electrochemical exfoliation process can produce a bulk amount of GN from graphite chunks. The anodic and cathodic reaction of graphite drives ions and radicals to intercalate into it and the formed gaseous species exerts a force thereby resulting in exfoliated GN sheets. The advantage of electrochemical exfoliation is its easy operation, as the reaction takes place via a single step [63].

Other exfoliation techniques are the sonication of GN precursors in solvents to overcome the van der Waals interactions of graphite. During sonication, the shear forces and cavitation, i.e., the formation of bubbles, and its collapse exert pressure on the bulk system, thereby resulting in the exfoliation. The solvent used should balance the intersheet attractive forces, i.e., it should minimize the interfacial tension. Solvents such as N,N-Dimethylformamide, N-methyl 2-pyrrolidone, and ortho-dichlorobenzene are some of the common solvents used [64].

### 4.2. Chemical Vapor Deposition

In the chemical vapor deposition (CVD) process, the pyrolysis of carbon precursors occurs in the hot zone of the reactor to form dissociated carbon radicals which are subsequently deposited on a metal substrate to form GN. The role of the metal is that of a catalyst as it lowers the reaction temperature, shortens the time, and also has an effect on the quality of GN produced [65]. Initially Cu and Ni substrates were used for the deposition, but later, many other transition metals were explored [66,67]. CVD can be of various types such as plasma-enhanced CVD, low-pressure CVD, or atmospheric pressure CVD. In plasma-enhanced CVD, the reaction occurs under a plasma, such as an argon plasma, which is an ionized material, and the ions subsequently transfer the energy to the chemical process as well as to the deposition process. The main advantage is that the deposition occurs at much lower temperature. The atmospheric pressure CVD is the normal CVD, and the reaction occurs at high pressure, while the low-pressure CVD uses heat to initiate the reaction of a precursor. The main advantage of the CVD process is that a controlled thickness of GN from a single to a few layers can be easily obtained [66,67].

### 4.3. Epitaxial Growth

In epitaxial growth, the atomic constituent of GN is deposited on substrates such as hexagonal boron nitride, silicon carbide, etc., using techniques such as nanolithography. This technique is highly precise and high-quality GN compatible with current semiconductor technology can be prepared by using this methodology. High-performance electronic devices, i.e., field-effect transistors, photodetectors, and chemical sensors have been successfully fabricated by this technique [67]. The high reaction temperature and high cost of the collector substrate are the main disadvantages associated with this process. The low-temperature pyrolysis of carbonaceous-GN-like carbon materials can be obtained by biomass (glucose, sucrose, gelatin, chitosan, rice husk, hemp, wheat straw, etc.) carbonization via a pyrolysis technique. Zhou et al. [68] showed that porous nitrogen-doped GN could be prepared from soyabean biomass via calcination and KOH activation. Similarly, Sun et al. [69] showed that pure and doped GN could be deposited on metal substrate at temperature as low as 800 °C. the advantage of this technique over CVD is that it eliminates the use of gaseous raw materials.

## 5. Antibacterial Potential of Graphene Nanostructure

Millions of people are infected with microbial infections, and it is becoming one of the most serious threats globally. Antimicrobial drugs such as antibacterial, antifungal, and antiviral drugs are commonly used to control infections. To analyze the antimicrobial activity of a nanoform of a material, a number of strategies are used, including the disk diffusion test, Agar well diffusion variant, Agar dilution method for the determination of minimum inhibitory concentration and minimum bactericidal concentration, colony forming units, and cell counting. Apart from these, DNA and ATP–based assays are also used for the determination of the antimicrobial efficacy of various nanomaterials including GBNs. Different antimicrobial responses have been seen based on the nature and properties of the bacterial species used as selection criteria [70,71]. The antibacterial activities of GN-based nanomaterials such as GN–carbonaceous materials composites, GN–metal composites, GN–metal oxide composites, and GN–polymer composites have been tested against many bacterial indicators, including *P. aeruginosa*, *E. coli*, *P. syringae*, *X. campestris*pv. *Enterococcus faecalis*, *Salmonella typhimurium*, *Streptococcus mutans*, *Fuscobacteriumnucleatum* and *Porphyromonasgingivalis*. The antimicrobial activities of GN and its composite have been well reported in literatures [1,2,3,4,70,71,72,73]. The nanoparticles of rGO have been described to be effective against *S. aureus*, *P. aeruginosa*, *Salmonella typhimurium*, *E. coli*, and *Enterococcus faecalis*, as well as other GN-based conjugated nanomaterials such as GO–AgNPs, which have been shown to be effective against *X. perforans*, and rGO–Cu against *E. coli* and *S. aureus* [37,72,73]. Moreover, metal-conjugated forms such as rGO–Cu have inhibited *P. aeruginosa* and *S. aureus*. Titanium dioxide, coupled with GO, has been reported to kill *E. coli* and *P. aeruginosa* [74]. The inhibitory effects observed with the conjugated form of nanomaterials are greater than those observed with the single constituent, which is due to an additive or synergistic effect. The synergistic antibacterial effect of GO–ZnO with low cytotoxicity has also been reported against *E. coli* with MIC 2.5–5.0 µg per mal [75]. The GO–Poly(N-vinylcarbazole) conjugated nanoform of nanomaterials significantly inhibited the biofilm formation in *R. opacus*, *E. coli*, *C. metallidurans*, and *B. subtilis*. The minimum inhibitory concentration of GQDs against *S. aureus* and *E. coli* was reported near 200 µg/mL [38,76,77].

### 5.1. Graphene–Carbonaceous Materials Composites

Since its invention in 2004, GN has been considered as a potential metal due to its excellent electrical, optical, and catalytic properties as well as its exceptional physical characteristics, such as mechanical strength and a large specific surface area. Moreover, with respect to other metal nanoforms, GN is much more readily available, affordable, and renewable [16,78]. Researchers worldwide have contributed their time to explore the utilities of GN against microbial pathogens, especially its interaction with bacterial cells. Various carbonaceous forms of GN such as colloidal sphere, microsphere, nanofibers, and nanotubes have been synthesized and investigated in different fields [5,7,8,9]. The first reports of GN’s antibacterial properties appeared in 2010. It has been studied as a film, coating, and as composite against *S. aureus* and *E. coli*, which are mainly killed by nanomaterials through membrane damage. Among GN, GO, RGO, graphite, graphite oxide, GO has been reported to have the most antibacterial potential and graphite oxide the least [2,3,79,80,81,82,83,84,85]. Moreover, traditionally used antibiotics, such as kanamycin, were described as less significant against bacteria compared to GN nanosheets [75,76]. GQDshave been reported to show a shape-dependent effect against *S. aureus*, e.g., cambered GQDs, with flat-shape GQDs having the least activity [76,77]. The structural integrity loss, chromosomal condensation, gene fragmentation, and inactivation of cellular enzymes are thought to be the primary causes of the carbon-dot-induced death of bacteria [24,25,78,79,80]. CNTs are typically of two types, i.e., single-walled and multiwalled carbon tubes. The significant antimicrobial activity was observed with single-walled nanotubes due to variation in size which could also be due to their large surface area [83,85]. The surface-to-volume ratio of carbon nanomaterials has an inverse relation with their size and it increases as their size decreases; this facilitates a stronger bond formation with the plasma membrane or cell wall of bacterial cells and more efficiently exerts their function. Due to the excellent surface properties of GBNs, they bind the bacterial contaminants of water and serve as a significant tool in the biological cleaning as well as metal cleaning of water [86,87]. They exert bactericidal as well as bacteriostatic actions targeting the cellular membrane and also the biofilm of pathogenic bacteria [76,77,78,79,81]. Their length and diameter also affect the antimicrobial potentialities and it increases with the decrease in their length (in solid) and diameter [80,81,82].

Fullerenes are enormous spheroidal molecules of carbon that have antibacterial capabilities against bacteria, namely, *E. coli*, *Streptococcus* spp., and *Salmonella*. The Gram-positive bacteria are more susceptible than the Gram-negative bacteria due to their cell wall complexity. Fullerenes have a bacterial inhibitory effect on Gram-negative bacteria such as *Pseudomonas putida* by altering the ratio of unsaturated fatty acids of the cell wall of *Pseudomonas putida* while raising the amount of cyclopropane fatty acids [82,84,88]. The antimicrobial effect of fullerenes on *E. coli* and *Shewanellaoneidensis* is dependent on the shape as well as electrostatic attraction. Fullerene C60-NH_2_ decreases the need for substrate, shows an improved efficacy of drugs for both Gram-negative and Gram-positive bacteria at a minimum concentration of 1 mg/mL, and destroys the integrity of bacterial cells [82,83,89].

### 5.2. Graphene–Metal Composites

To enhance the antibacterial properties of GBNs, their composites with Ag have been extensively studied and their composites with other metals and metal oxides have also been reported. These metals include Fe, Zn, and Cu and their respective oxides. The atoms are conjugated by covalent linkages, van der Waals forces, electrostatic interactions, etc. The optimum size of nanostructure of Ag for an antimicrobial activity has been observed to be 20 nm in Ag/rGO composites compared to too-small and large-sized particles [1,2,3,4,90,91,92,93,94]. The incorporation of Fe is used to produce magnetic GN which has been reported to be 100% bactericidal against *E. coli*. This is due to variable superparamagnetic characteristics, a large number of adsorption sites, ease of separation and adsorption of heavy metals such as Cr, Pb, and As [90,91,92,93,94]. The GO in conjugation with different atoms have been reported to possess a distinct antimicrobial activity. The Zn conjugate possessed a maximum antimicrobial activity while the least activity was reported with steel [1,2,3,4]. The decreasing order of antimicrobial activity, i.e., the antimicrobial potential against bacteria as well as fungi of the combination of different atoms with GO is described as: GO–Zn > GO–Ni > GO–Sn > GO–Steel, GO–Zn, GO–Cu, GO–Mn and GO–Se. It is interesting to note that the antibacterial effects of the selenium nanoparticle composites on Gram-positive and Gram-negative bacteria (*S. aureus*, MRSA, *E. coli*) were different. *S. aureus* and MRSA, two types of Gram-positive bacteria, were particularly susceptible to the impacts of composites on their bacterial cultures. However, only at the maximum applied dose did the same composites affect Gram-negative bacteria, specifically *E. coli*. At 25 °C and a 25 mg/mL concentration [91,92,93], Kurantowicz et al. [95], reported the strong efficacy of GO and rGO for inhibiting the bacterial pathogens *S. enterica* and *Listeria monocytogenes* [95]. Radiation-exposed (758 nm) nanoforms of Fe-GO, TiO_2_-Au–rGO, and Au-GO, have been reported to have broad spectrum antibacterial activities against Gram-negative, Gram-positive, and fungal pathogens [96,97]. Asa matter of fact, the combination of NPs with GO or rGO have resulted in the development of new nanocomposites that have represented an additive inhibitory potential against many pathogens [98].

### 5.3. Antimicrobial Potential of Graphene–Metal Oxide Composites

Kavitha et al. [43] discovered that the nanostructure of ZnO/GN nanosheets had much higher antibacterial efficacy compared to pure GN, which was produced using a zinc benzoate dihydrazinate complex as a single-source precursor at 200 °C. Wang et al. reported an increase in the antibacterial activity of ZnO–GO nanosheets synthesized at low temperature using dimethylformamide. A GO/ZnO nanocomposite synthesized by Wang et al. [91] showed effective growth inhibitory properties against *E. coli* [89,90,91]. Ag–GO and ZnO–GO were tested for their antibacterial efficacy against *Klebsiella pneumonia*, *Enterococcus faecium*, *E. coli,* and *S. aureus*. According to Whitehead et al., Ag–GO had a greater antibacterial effect than ZnO–GO. The MIC for *E. coli/E. faecium* was higher (0.125 mg/mL) as compared to *S. aureus/K. pneumoniae* which was 0.25 mg/mL. They discovered that adding Ag to GO increased the activity of AgNPs against bacteria [2,3,4,99,100,101].

The rGO/iron oxide effectively destroys pathogens by producing heat locally and a significant quantity of hydroxyl radicals [24,25,102]. The Cu metal and its nanostructure have been reported in many applications such as hydrolysis, hydrogen gas sensors, and solar photovoltaic cells. This metal with GO has potential antioxidant and antibacterial activity. The IC_50_ against epidermoid cancer cell was 44.86 ± 1.74 μg/mL [9]. Moreover, hydrothermally synthesized rGO–CuO nanocomposite films has significantly inhibited the growth of *Pseudomonas aeruginosa PAO1* [94,95]. It was possible to create a unique nanocomposite film containing bacterial cellulose and GO–CuO nanohybrids that was active against Gram-positive bacterial pathogens due to the ROS–generated membrane damage of the tested bacterial strains [22,23,24,72].

*Staphylococcus aureus* and *Pseudomonas aeruginosa* were used as indicator strains of Gram-positive and Gram-negative bacteria, respectively, to examine the antibacterial activity using plate counting and optical density (OD_600_) techniques [70]. The findings reported that SnO_2_@GN had a 1–3 fold higher cellular toxicity than GN alone for the growth of *S. aureus* and *P. aeruginosa*. Due to the synergistic impact, SnO_2_@GN’s cytotoxicity was 3.6 times higher against *Pseudomonas aeruginosa* [93]. Similarly, cobalt oxide nanoparticles/rGO have also been reported as significant antibacterial composite [94]. According to Paek et al., gold NPs alone have growth inhibitory capabilities for *E. coli*, but when combined with GO or rGO, they become more efficient at inhibiting the growth of bacteria such as *S. aureus* and *B. subtilis* [103].

### 5.4. Antimicrobial Potential of Graphene–Polymer Composites

A viable method for creating new antibacterial nanostructures is the integration of a carbon-containing complex into polymeric composites. Material-enriched nanofillers serving as reinforcement and polymers used as a matrix are known as polymeric nanocomposites [104,105]. Novel antifungal or antibacterial polymer-based nanocomplexes with synergistic antimicrobial potentials can be made by combining GN of various shapes or sizes with polymers for various desirable properties [51,52,53,54]. Many GN–based nanocomposites are synthesized using polymers such as biodegradable synthetic polymers, natural polymers, methacrylic, and acrylicpolymers [104]. They offer many significant properties and have been explored for human welfare in different fields including diagnostics and therapeutics [99]. Several methods such as solution blending, melt compounding, in situ polymerization, electro polymerization, and latex mixing can be used to create polymer-based nanomaterials with GN [104,105,106,107,108,109]. The properties of composite depend on the preparation technique of the nanocomposite, the concentration of the nanofiller, and the degree of dispersion and morphology of the nanofiller into the matrix.

The antimicrobial potentials of methyl methacrylic acid (MAA) and acrylic acid (AA) based polymers have been explored as good antibacterial agent in a complex with GBNs. Several researchers have shown that GN addition to AA and MAA produced complex polymers with significantly enhanced antimicrobial properties. The reduced activity of silver nanoparticles due to the agglomeration, was overcome in GO (1–2 w%)-Ag-PMMA polymeric composite sheet, and the sheet showed much higher antibacterial activities against *S. aureus*, *S. mutans*, and *E. coli* due to them wrapping of bacterial cells by sheets [106]. Moreover, pressured gyration was used to generate PMMA fiber meshes enriched GO (8 w%), and the significant bactericidal efficacy of these matrices against *Escherichia coli* was studied [100]. The increased antibacterial activity was observed through the cross-linking of the Ag/GN nanostructure, which was cross-linked with acrylic acid and N,N′-methylene bisacrylamide at various mass ratios to create hydrogels for *E. coli* and *S. aureus* [104,105]. An excellent biocompatibility, good extensibility, high swelling ratio, and high antibacterial activity were all displayed by the hydrogel with the ideal Ag:GN mass ratio of 5:1 [101,102]. PNIPAM hydrogels enriched with GO and GO/CNT nanocomposites were evaluated against *P. aeruginosa*. Poly(N-vinylcarbazole)-based biocompatible synthetic materials was also developed against Gram-positive bacteria *B. subtilis* and *R. opacus* as well as Gram-negative bacteria *E. coli.* [104,105,110,111,112,113]. The increased antibacterial activity against the bacteria was higher with the polymer complex than with GO which could be due to the encapsulation of bacteria by the hydrogel. PVA, or poly (vinyl alcohol), is a water-soluble synthetic polymer that has a great potential as a low cytotoxic, biodegradable, and biocompatible material for use in a variety of commercial, medical, and industrial applications [112,113]. GO was added to a nanocomplex in concentrations of 1, 5, and 10 weight percent using a solution-casting technique, and its antibacterial effects on *S. aureus* and *E. coli* were evaluated and applied in beverage packaging and as a film in food preservation to inhibit the Gram-positive bacteria [103]. Moreover, PVA matrix composites with Ag nanoparticles anchored to GO by a solution-casting process are currently synthesized using a one-step chemical reduction process. Due to their additive antibacterial activity, the nanomaterials displayed highly efficient inhibitory activities against *E. coli* and *S. aureus* [104,105]. Moreover, GO–chitosan–TiO_2_ has also been described as having a food-preservative action [114,115]. Although PLA has significant drawbacks, such as poor barrier qualities, itis one of the most commonly utilized biodegradable materials for medical purposes. It has been blended with other GN-based nanostructures and those materials combinations which offer increased inhibitory potential against *S. aureus* and *E. coli.* PLA–based composites such as GO–ZnO–PLA and GO/Ag–PLA have been synthesized and evaluated as potential antimicrobials [116]. Although PVDF suffers considerably from befouling, itis a biocompatible, nonbiodegradable, flexible, and reasonably cost-effective polymer and is commonly utilized in water purification systems. The creation of PVDF-enriched nanocomposite membranes with antifouling capabilities which was achieved by incorporating nanoparticles has received a lot of attention from researchers working in the field of food technology. Ag nano hybrid can be added to PVDF–GO films to increase their antibacterial resistance. In the electrospun PVDF–Ag–GO fiber mats, the PVDF membrane itself facilitates the bacterial adhering to membrane while the Ag-enriched PVDF–Ag–GO nanocomposite released free Ag ions to inactivate the bacterial enzyme involved in DNA synthesis [116,117,118]. Although inactive against bacteria, a synthetic and biodegradable polymer, PCL is another choice of filler for nanocomposite fabrication. However, when it is coupled with rGO/Ag, it exhibits biocidal action [119,120]. The other biodegradable polymer polyethylene glycol has been used to synthesize a PEG/GO/Ag nanocomposite to control the food spoiling due to Gram-positive and Gram-negative bacterial population mainly *E. coli* and *S. aureus*. PVA, PEG, and fumaric acid based polyester have recently drawn a great deal of interest for use in the medical field due to their excellent biodegradability and biocompatibility [121,122,123,124,125,126].

## 6. Mechanism of Action of Graphene Nanomaterials

Different forms of GN and GBNs have been reported with variable efficacy delivery, and modes of action against a wide range of microbes, which depend on the form of the nanoparticles, their features, and the nature of microbes [13,14,15,16,24]. It has been demonstrated that the effective and nontoxic concentration of the GN nanoform is 50 mg/L. The minimum inhibitory concentrations of the nanocomplex of GN (Gr/CS/Fe_3_O_4_ NCs) against *K. pneumoniae* and *P. aeruginosa* have been reported to be 70 and 60 g/mL [127]. The AgNPs–GN composite was reported to be more potent than pure AgNO_3_at very low doses of 5 g/mL [128,129,130]. The MBC value of 20 g/mL, which was less compared to that of GN/CS/Fe_3_O_4_ NCs, indicates the more efficacious nature of AgNPs–GE. The GO flakes have a high effectivity against *P. aeruginosa* and *S. aureus* in the range of 4–128 g/mL of GO concentration. Moreover, for Ag/rGO, the MIC was 50 g/mL and the minimum bactericidal concentration values were 400 g/mL and 50 g/mL for *Pseudomonas aeruginosa* and *S aureus*, respectively [117,118]. Recent research has shown that GN has interesting properties to control bacterial pathogens. Different mechanisms such as membrane stress, including oxidative stress, and electron transfer, have been hypothesized for GN’s bacteriostatic or bactericidal action. The direct contact of GN with the bacterial membranes can physically harm the membranes [128,129,130,131,132]. The antibacterial mechanisms of action of GN-based nanomaterial are shown in Figure 5. Moreover, the GN noncomplex has also been reported to exert their significant inhibitory action on many pathogenic fungi.

### 6.1. Antibacterial Activity of Graphene-Based Nanomaterials

The most commonly proposed antibacterial mechanisms of GN and its nanoparticles are due to oxidative stress induction, membrane damage, protein dysfunction, and transcriptional arrest. The frequent use of these GN-based drugs has imparted microbial resistance and thus the urgent need of alternatives that overcomes this resistance has accelerated the discovery of new, safe, and effective nanomolecules. GN-based nanomaterials have been tested against microbial infections including COVID-19 [30]. They have also been used in conjugation with preexisting approved drugs and polymers to enhance the delivery and efficacy of drugs in microbial infection. GN-based green nanotechnology has shown broad-spectrum antibacterial activities. Moreover, a limited cytotoxicity and resistance of bacteria to GN have been reported [121,122,123,124,125,126].Physical and chemical antibacterial mechanisms of action, such as physical damage, membrane destruction, and interaction with the lipid-like biomolecules of microbial cell damage, are commonly reported. Chemical methods mediate killing action on bacterial cells through charge transfer or the cellular production of ROS [24,25]. Photothermal ablation has also been reported as an antibacterial route to control pathogens as described by Wu et al. [132]. An electron microscopic examination has confirmed the cell membrane damage of *E. coli* by GN-based fabricated nanomaterials. GN nanowalls have been reported to kill bacteria through the bursting of cell membranes, which facilitates the cellular transport of some life-supporting biomolecules towards the exterior of the cells [120,121,122,123,124,125,126]. The efflux of the enzyme glycosidase has been reported due to a loss of cell membrane integrity caused by nanowalls in a bacterial strain of *E. coli*. Moreover, the density of nanomaterial that surrounds the bacterial cell is also considered a key factor responsible for the bactericidal activity. The density of the nanosheet of GN has facilitated the pore formation in the cell membrane of the bacteria [2,3,4,24,25].

### 6.2. Antifungal Activity of Graphene-Based Nanomaterials

Nanotechnology has been implemented to increase the effectiveness of pathogen-controlling potentialities in the area of agriculture and can be employed as the current potential techniques against phytopathogens [133,134,135]. Nanomycostatic and mycocidal formulations with a low toxicity, more water solubility, faster transport, and greater bioavailability and efficacy have been described. Nanotechnology can increase therapeutic efficiency in an agricultural environment and can change the current methods that are used to control plant pathogens [134]. The formulation of nanofungicides may provide some strength, such as an improved performance and bioavailability of fungicides, reducing toxins, and increasing the solubility in water, can guide the delivery of active ingredients and improve their shelf life [134,135,136].

Different types of nanoparticles and other types of nanomaterials, such as agronanofungicides, *Z. multiflora*, and ginger essential oil nanoformulations, have been reported to be effective and safe in controlling pathogenic fungi in plants. Due to their crystalline structure, resulting in possible drug expulsions due to the crystallization process, there is a need to enlighten the formulations at the most appropriate temperature to store them. Mesoporous silica nanoparticles with a good charge in place can provide a significant cytotoxic effect when compared to anion and neutral species [134,135]. Agriculture plays a crucial role in providing food and is a source of income in many countries. It is a major source of livelihood for people in rural areas; about 86% of rural people rely on agriculture. About 15–18% of crop losses occur due to pests, while weeds and pathogens cause 34 and 16% of losses, respectively. Fungal infections cause 70–80% of yield losses [134]. Approximately 1.5 million species are classified under the “fungus” regime, and these fungal organisms are usually parasitic and naturally saprophytic, causing various diseases in agricultural crops. Every year, fungal infections can cause significant crop yield losses around the world [133,134,135,136,137] (Figure 6). Currently, disease control depends on the use of agricultural chemicals, for example, fungicides. In addition to the many positive benefits, such as a rapid action, reliability, and high availability, fungicides can have a detrimental effect on unintended organisms due to their toxicity and systemic action by disrupting metabolite levels in the biosynthetic pathway of odors. The development of amino acids within soil microorganisms, the development of resistance, and the re-emergence of insects in the environment are frequently seen. Furthermore, it is estimated that 80–90% of sprayed fungicides are lost to the environment after or during application.

GBNs have broad-spectrum mycocidal and mycostatic activities against many pathogenic fungi [133,134]. The spore inhibitory activities of GN nanomaterials have also been explored. The antifungal potential of GO against fungal pathogens has been widely reported in various ways [134,137,138,139,140], mainly by GO nanosheet aggregation and cell membrane disruption, which results in the reduction of membrane potential and the cellular leakage of electrolytes through the membrane (Figure 7). Moreover, GO offers a significant antibacterial effect due to its high photothermal efficacy. The dose-dependent antifungal activity of GO against *S. cerevisiae* has been evaluated up to a range of 0 to 600 mg/mL [137]. The antifungal mechanism has been reported to increase ROS generation and reduce mitochondrial transmembrane potential, which results in an affected expression of apoptosis-related genes, such as SOD, Yca1, Nma111, and Nuc1 [137]. In the dose-dependent antifungal studies on GO, the disruption of the fiber structure of the tested white rot fungi, *P. chrysosporium* was reported at a high concentration of 4 mg/mL, while at a low dose, it reduced the pH of the medium and stimulated the growth of the fungal strain. Xie et al. [95] exposed *P. chrysosporium*, a white rot fungus, to GO concentrations of 0–4 mg/mL for 7 days. Their results showed that low concentrations of GO stimulated the cells growth and caused more acidic pH values of the culture media. In addition, the scanning electron microscopy images demonstrated that GO induced the disruption of the fiber structure of *P. chrysosporium*, where some very long and thick fibers were formed at 4 mg/mL [138]. The fungal strains such as *A. niger*, *A. oryzae*, and *F. oxysporum* were significantly inhibited by rGO at a concentration of 0–500 g/mL. Among these fungal pathogens, *F. oxysporum* was inhibited potentially more (IC; 50 g/mL) than the strains *A. niger* and *A. oryzae* (IC50; 100 g/mL). The antifungal mechanism facilitated by rGO nanosheets is due to the interaction of nanosheets with chitin and polysaccharides components of the cellular membrane. Subsequently, the ROS generation of rGO nanosheets mediates the interaction of chemical groups of chitin and other polysaccharides on the cell wall of fungi. The antibacterial and antifungal mechanisms of action of GO explored in other studies [141] showed that the mechanism of action is through the disruption of the cell membrane of. *P. syringae*, *F. oxysporum*, *X. campestris*pv. *Undulosa*, and *F. graminearum*. The GO nanoform has been reported to inhibit up to 90% of the tested fungal strains while it swells and lyses the macroconidia up to 80% [132,134]. The synergistic effect of GO and other nanoparticles especially Ag has been extensively explored to make effective antimicrobial products. To increase the antifungal activity of carbon nanoscrolls, they were supplemented with silver nanoparticles (AgNPs) and compared to the antifungal activity of GO–AgNPs and composites [140]. Carbon nanoscrolls–AgNPs have shown a long-term activity against *C. albicans* and *C. tropical* compared to GO–AgNPs and composites. GO itself alone has not been reported to have an inhibitory effect, while the GO–AgNPs composite has a potential fungal inhibitory effect, which has more potential against *C. albicans* than *C. tropical* [136,142]. Drug resistance and herbicide, weedicide, and pesticide resistance have also been seen in plant pathogens including bacteria and fungi [1,2]. To control this resistance towards plant pathogens, many combinations of drugs and nanomaterials have been tested and well described with a broad antimicrobial spectrum that acts through a unique mechanism [4,5,6]. The synergistic effect of currently used drugs with many antifungal approaches has been applied as a significant approach to control plant pathogens. The fungicidal effect of GO nanomaterial with antifungal drugs *Mancozeb*, *Cyproconazol*, and *Difenoconazole* on the spore germination, mycelial biomass, and mycelial growth of *F. graminearum* was more significant compared to when it was used alone [135,136].

## 7. Toxicity of Graphene-Based Nanomaterials

The excellent surface properties, presence of functional groups, dispersion, stability, including administration and delivery, extended time, and simple way of synthesis arethe basis of exploration of GN nanoscience in various biological and biomedical applications [13,14,15,16,33,34]. However, after synthesis, care should be taken to avoid carbonaceous debris, the incorporation of several metallic impurities, and interruptions in the GN structure as these may produce toxicity [33,34]. These factors could result in varying toxicity responses [143,144,145,146,147]. Several reports have recently investigated the in vitro toxicity due to cellular accumulation of many nanostructures of GN, including GQDs, in microbial as well as in various types of mammalian cells. Overall, these investigations concur that the nanostructure of GN derivatives has a low toxicity and good cellular absorption, making them suitable for biomedical applications [33,143]. Many experts, however, argue that cytotoxicity of micro/nanosized GN is far more significant than that of GQDs, and that it should not be overlooked in biological research. [42,43]. Several studies conducted on GO-based nanoflakes and GOS have been linked to serious toxicities and pulmonary illnesses. Other studies conducted extensively [49] have revealed the varied toxicity of GOs to specific cells is governed by its surface properties. Overall, the mortality of GN nanomaterials is highly related to the size of the particles, which could explain why a few nanometer-sized GQDs are less harmful than nanosized GOs [20,21,33,34,35,36,37,38,39]. On the other hand, Akhavanet al. claimed that the size of the nanoparticle has no relation to the toxicity of GN-based products. However, the interaction of GN sharp edges with living cells provides more probable empirical evidence regarding toxicity. This relates to the researchers’ second argument, that indeed nanosized GOs could harm cell lines. As a result, the toxicological processes relating to the shape, size, and function of GN nanoforms are unknown thus necessitating further research. Numerous research groups have also looked at the impact of edge functionalization on cell permeability and toxicity. The covalent binding of hydrophilic molecules, such as PEGs, to the edges of GN has long been thought to improve the biocompatibility and solubility of GN in biological contexts [9,16]. The effects of adding new structural features to GQDs, such as –COOH, NH_2_, CO–N(CH_3_)_2_, and –PEG, on cytotoxicity have also been studied [34,39]. An approach for the data analysis was used, which revealed no significant differences in toxicity in such GQD variants. However, the membrane permeability improved in the following order: PEG, OH, and NH_2_ [49]. These findings are promising for investigators who want to use improved GN derivatives due to their low cytotoxicity. Sasidharan et al. 2011 discovered that virgin GN(hydrophobic) and carboxylated GN (hydrophilic) have different effects in biological settings [141]. In contrast to pure GN, carboxylated GN exhibits impaired interactions with the cell membrane, resulting in negative impacts including cellular membrane distortion, elevated intracellular ROS levels, and apoptosis [24,25]. Edge surface modification is important in cell–nanoparticle interactions as well as in peroxidase enzyme assisted biodegradation. The in vivo toxicity, bioavailability, or clearance of GN–based nanomaterials is determined by the nanomaterial’s properties, dose, route of administration, time of exposure, and animal used. The intraperitoneal administration of 4 mg/kg body weight for 4 weeks of GN and GO has concentrated in connective and lipid tissue in the proximity of the liver and spleen serosa [143]. Liu et al. reported in their study that highly accumulated amounts of GN nanoparticles is problematic for the kidney. Pulmonary thromboembolism has been reported with GO, which could be due to the high charge and large surface area. However, because they are small and are not heavily accumulated, they are easily removed by the kidney’s glomerular filtration [144]. Following intravenous injection, PEGylated NGS largely builds up in the reticuloendothelial system, which includes the spleen and liver, and can be subsequently eliminated, most likely via both fecal and renal excretion [147]. A hematological testing, blood biochemistry, and histological evaluations of the treated mice reported that PEGylated NGS did not result in noticeable toxicity at the given dose of 20 mg/kg. Similarly, the GO-PVP nanocomplex was accumulated in the spleen, lungs, and liver, which was then eliminated in urine after entering the deep suborgans of the lung, liver, and spleen, where GO could then be transferred and cleared in the intraorgans. The deformation property of GO facilitated its transfer across the continuous, discontinuous, and continuous fenestrated endothelium. No undesirable toxicity such as pulmonary thromboembolism or platelet effects was produced by the use of the amine group (NH_2_-GO) as the GO caused a 46% blockage of the pulmonary blood arteries [145].

## 8. Conclusions

Due to their unique features, such as a high surface-to-volume ratio, mechanical flexibility, and thermal stability, GBNs have been studied extensively for a wide variety of applications over the last decade. GBNs have shown potential applications in catalysis, solar cells, biosensors, drug and genetic delivery, imaging, photothermal therapy, tissue engineering, and stem cell technologies. This report outlined the recent improvements in the understanding of the antimicrobial potential of GBNs, focusing primarily on the antifungal and antibacterial effects of GBNs against different bacteria and fungi. The antibacterial ability of GN-based nanostructures has been evaluated against pathogenic bacteria, fungi, and viruses, either alone or in conjugated forms. They have been proven to be safe preservatives and can be a potential replacement for harmful chemical preservatives. One of the hurdles for GN-based materials in biological systems is their dispersibility, while the high specific surface area, biocompatibility, good diffraction strength, high Young’s modulus, quick ionic migration, and high electrical and thermal conductivities are all its advantageous features. Antibacterial, antifungal, and antiviral medicines are routinely used to treat infections. *P. aeruginosa*, *E. coli*, *Enterococcus faecalis*, *Salmonella typhimurium*, and *S. aureus* have all been demonstrated to be susceptible to GN-based nanomaterials. *E. coli* has been shown to be killed by GO-TiO_2_composite. In *R. opacus*, *E. coli*, *C. metallidurans*, and *B. subtilis*, the GO-Poly(N-vinylcarbazole) conjugated nanoform of nanomaterials greatly prevented biofilm formation. To control plant pathogens, many combinations of drugs and nanomaterials have been described with a broad antimicrobial spectrum that acts through a unique mechanism. The synergistic effect of currently used drugs with many antifungal approaches has been applied as a significant approach to control plant pathogens.

## Figures and Tables

**Figure 1 nanomaterials-12-04002-f001:**
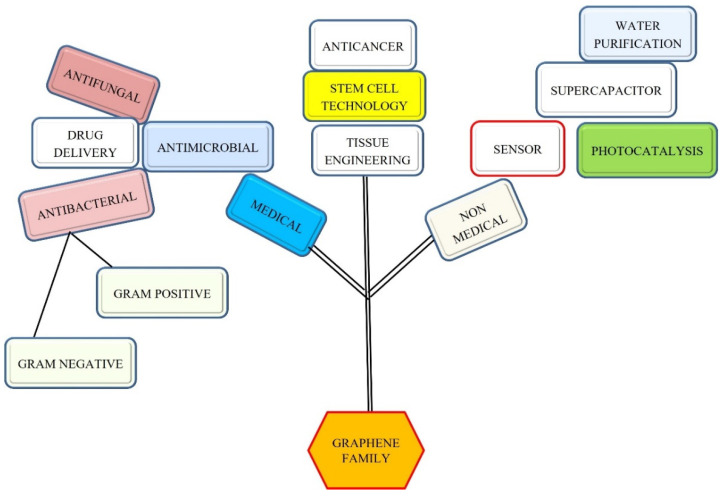
Potential applications of GN-based nanomaterials in different fields.

**Figure 2 nanomaterials-12-04002-f002:**
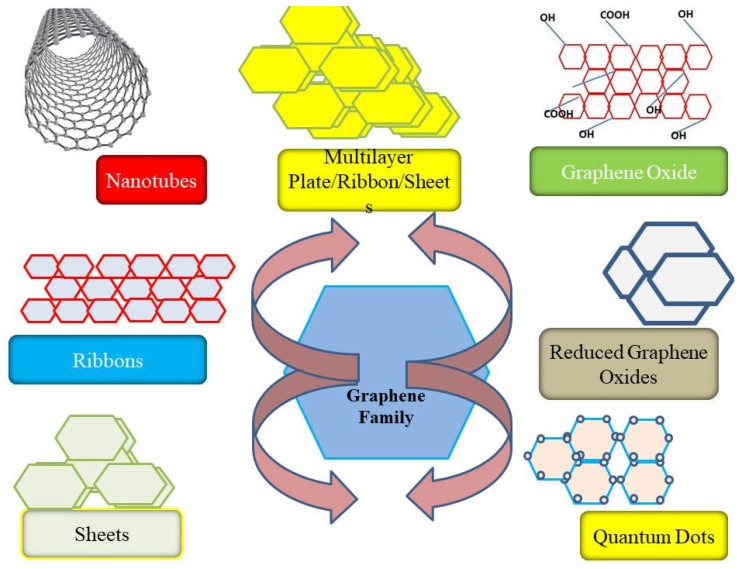
Different forms of GN-based nanomaterial.

**Figure 3 nanomaterials-12-04002-f003:**
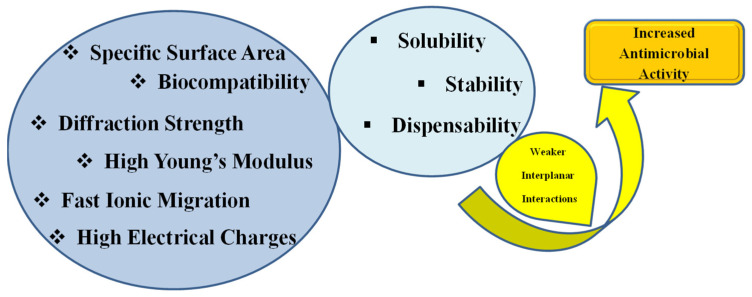
Properties of GN materials for antibacterial activities.

**Figure 4 nanomaterials-12-04002-f004:**
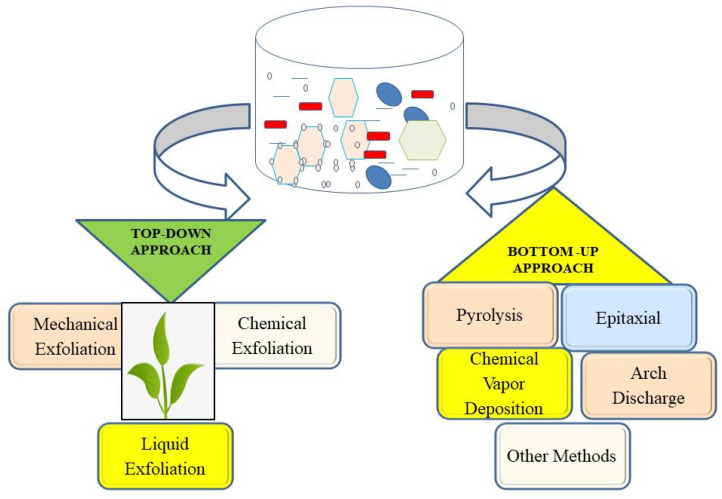
Synthesis methods for GN-based nanomaterials.

**Figure 5 nanomaterials-12-04002-f005:**
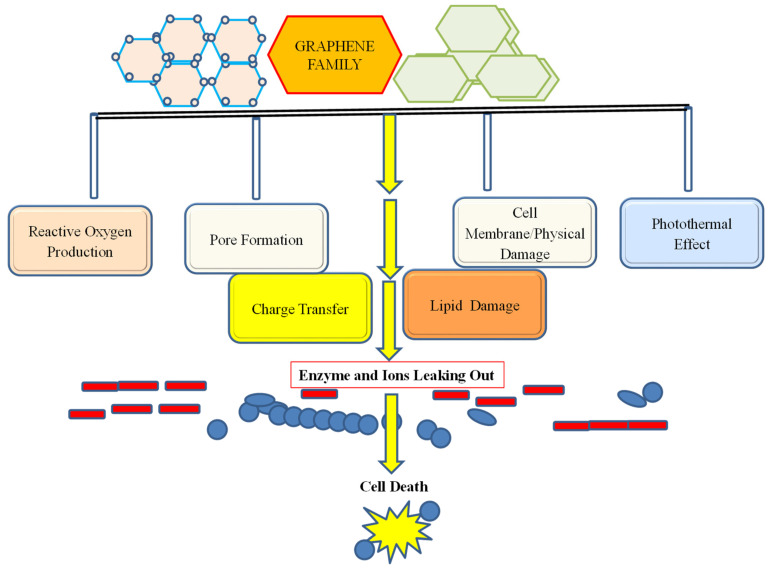
The antimicrobial action exerted by a GN complex to control the growth of bacteria and fungi which are mainly reported as reactive oxygen formation, membrane damage/pore formation, photothermal effect, and charge transfer.

**Figure 6 nanomaterials-12-04002-f006:**
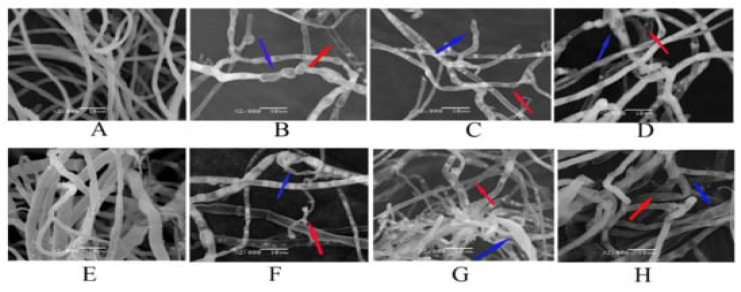
SEM images of the structural changes in the morphology of *F. graminearum* induced by fungicides, GO–fungicides and GO. Mycelia of the control (**A**), GO–treated ((**E**), 100 µg/mL^−1^), Man–treated ((**B**), 20 µg mL^−1^), Man–GO–treated ((**F**), 20 µg mL^−1^), Cyp–treated ((**C**), 100 µg mL^−1^), Cyp–GO–treated ((**G**), 100 µg mL^−1^), Dif–treated ((**D**), 200 µg mL^−1^) and Dif–GO–treated ((**H**), 200 µg mL^−1^) *F. graminearum*, respectively. The position marked by a red arrow indicates hollowness, and that marked by a blue arrow indicates swelling [134].

**Figure 7 nanomaterials-12-04002-f007:**
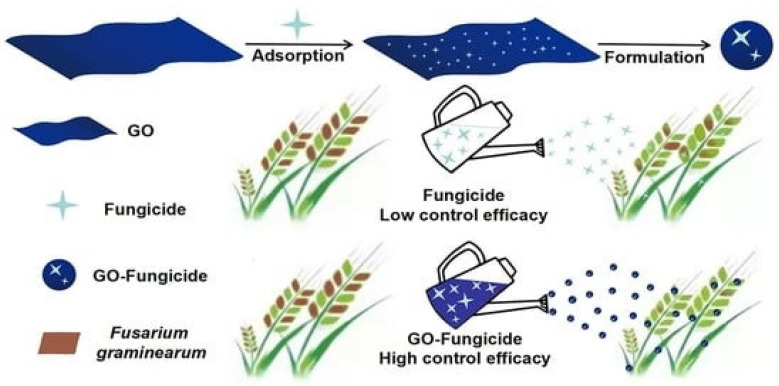
Schematic representaion of GO based antifungal formulation and its application in plant protection [134].

## Data Availability

Not applicable.

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
