# Peer review of "Antimicrobial Activity of Graphene-Based Nanocomposites: Synthesis, Characterization, and Their Applications for Human Welfare"

_nanomaterials, 2022, doi:10.3390/nano12224002_

Round 1
Reviewer 1 Report
Review article entitled “Antimicrobial activity of graphene based nanocomposites: synthesis, characterization, and their applications for human welfare” submitted to Nanomaterials, MDPI can be published after following corrections. However, in the present form it cannot be published
1. Abstract is not fully aligned to the manuscript.
2. Literature survey is very poor because it doesn’t explore recently published works related to graphene. Please see following (https://doi.org/10.1021/jacs.5b11411, https://doi.org/10.1016/j.jsamd.2020.01.006, https://doi.org/10.1080/10408436.2015.1127206, ) references relate to graphene. These citations will be very important for the readers especially for the beginners working on antimicrobial properties of graphene.
3. What is the reason for not discussing antimicrobial properties of different nanomaterials in comparison to graphene ?
4. Efficiency and working mechanism of graphene and related materials for antimicrobial action needed to be discussed.
5. Nearly 25% of references are unwanted or not very relevant.
6. The reference style is not identical.
7. There is no information about the copyrights of the figures. It needs confirmation.
Author Response
Dear Reviewer,
Thanks for your valuable comments. The suggestions have been incorporated accordingly
Comment-1 Abstract is not fully aligned to the manuscript.
Responce-1 Thanks for your valuable comments, and the abstract has been refined.
Comment-2 Literature survey is very poor because it doesn’t explore recently published works related to graphene. Please see followinghttps://doi.org/10.1021/jacs.5b11411https……………
Responce-2. The text has been upgraded with the addition of new references and as per reviewer suggestion.
Comment-3 What is the reason for not discussing antimicrobial properties of different nanomaterials in comparison to graphene ?
Response -3 The justification for describing the antimicrobial properties for graphene have been added.
Comment-4 Efficiency and working mechanism of graphene and related materials for antimicrobial action needed to be discussed.
Response -4 The suggestions have been incorporated
Comment-5 Nearly 25% of references are unwanted or not very relevant.
Response -5. The references have been updated as per reviewer suggestions.
Comment-6 The reference style is not identical.
Response -6 The required correction has been incorporated
Comment-7 There is no information about the copyrights of the figures. It needs confirmation.
Response -7: The figures used in this manuscript are self-created and used one referenced.
Reviewer 2 Report
In the manuscript nanomaterials-1988991, the authors have conducted a review synthesis, antimicrobial potential, and biological toxicity of various graphene nanomaterials.
Conclusion: Provided that the authors can address the comments below, this manuscript may be published in Nanomaterials.
Major comments:
- It is unclear how graphene nanomaterial is used in applications. Is it applied in vivo or in vitro? What are some examples of applications in biological systems?
- When toxicity is discussed, it focuses on cellular toxicity (cytotoxicity). In the case of potential in vivo use of the graphene nanomaterial, are there any concerns of systemic toxicity? How instance, how is the nanomaterial cleared out of a human subject’s system? Is there any potential damage to the kidneys, liver, or spleen? These considerations could be briefly discussed in the toxicity section for a more complete view.
- There are a lot of incomplete and/or grammatically incorrect sentences in this manuscript. A thorough review of the language and grammar used in the manuscript should be done before publication. See comments below for some instances (this is not a complete list of all the mistakes).
Minor comments:
- The sentence on line 42 is missing a period.
- The sentence on lines 44-45 is incomplete.
- The sentence on like 596 is incomplete.
- All Latin terms in the manuscript should be made italics (e.g., lines 626-627). A mixture of italics and regular font is found throughout the manuscript.
Author Response
Dear Reviewer thanks for comments and suggestions have been incorporated as highlighted in the manuscript
Major comments:
Comment-1 It is unclear how graphene nanomaterial is used in applications. Is it applied in vivo or in vitro? What are some examples of applications in biological systems?
Response -1. The IN-Vitro, applications of graphene are extensively described and a brief about In Vivo are also described.
Comment-2 When toxicity is discussed, it focuses on cellular toxicity (cytotoxicity). In the case of potential in vivo use of the graphene nanomaterial, are there any concerns of systemic toxicity? How
Response -2. The suggestions have been incorporated.
Comment-3 instance, how is the nanomaterial cleared out of a human subject’s system? Is there any potential damage to the kidneys, liver, or spleen? These considerations could be briefly discussed in the toxicity section for a more complete view.
Response -3. The suggestions have been incorporated
Comment-4 There are a lot of incomplete and/or grammatically incorrect sentences in this manuscript. A thorough review of the language and grammar used in the manuscript should be done before publication. See comments below for some instances (this is not a complete list of all the mistakes).
Response -5. The corrections have been incorporated
Minor comments:
Comment-1 The sentence on line 42 is missing a period.
Response -1. The corrections have been incorporated
Comment-2 The sentence on lines 44-45 is incomplete.
Response -2. The corrections have been incorporated
Comment-3 The sentence on like 596 is incomplete.
Response -3. The corrections have been incorporated
Comments-4 All Latin terms in the manuscript should be made italics (e.g., lines 626-627). A mixture of italics and regular font is found throughout the manuscript.
Response-4. The corrections have been incorporated
Reviewer 3 Report
The paper is devoted to review antimicrobial potentialities of graphene nanomaterials. The topic is generally interesting, however the paper contains unexplained places (below) and need major revisions.
1) In introduction should be added more relevant references. The reference list should be extended with publications in the last years (2019-2022).
2) All abbreviations should be explained by first using. For example, IG line 69.
3) Please add pictures related with the main topic of the paper. Currently, only Fig. 5 is strictly related with antimicrobial properties of graphene based nanomaterials.
4) Conclusions should be rewritten in more informative way.
5)The specific applications area of antimicrobials properties of graphene based nanomaterials should be indicated in the paper text.
6) It would be nice to review methods of investigations of antimicrobial properties of graphene based nanomaterials.
Author Response
Dear reviewer, Thanks for your valuable comments on manuscript and suggestions have been incorporated as highlighted.
- In introduction should be added more relevant references. The reference list should be extended with publications in the last years (2019-2022).
Responces-1: The corrections have been incorporated
- All abbreviations should be explained by first using. For example, IG line 69.
Responces-2: The corrections related to abbreviations have been incorporated.
- Please add pictures related with the main topic of the paper. Currently, only Fig. 5 is strictly related with antimicrobial properties of graphene based nanomaterials.
Responces-3: The suggestion has been updated.
- Conclusions should be rewritten in more informative way.
Responces-4: The conclusions have been refined and improved..
5)The specific applications area of antimicrobials properties of graphene based nanomaterials should be indicated in the paper text.
Responces-4: The invitro and in vivo activities have been added in the manuscript.
6) It would be nice to review methods of investigations of antimicrobial properties of graphene based nanomaterial.
Round 2
Reviewer 3 Report
Authors make proper corrections according to reviewer remarks and I suggest to publish the paper as it is.
Author Response
Dear Reviewer,
Thanks for your valuable comments and changes have been inserted accordingly.
The whole manuscript has been revised and corrections have been incorporated.